# Theoretical Study on Vibrationally Resolved Electronic Spectra of Chiral Nanographenes

**DOI:** 10.3390/molecules29173999

**Published:** 2024-08-23

**Authors:** Yijian Ma, Xian Feng, Wenxiong Yu, Chengshuo Shen

**Affiliations:** School of Chemistry and Chemical Engineering, Zhejiang Sci-Tech University, Hangzhou 310000, China; 2023211001030@mails.zstu.edu.cn (Y.M.); 202220103105@mails.zstu.edu.cn (X.F.); 202230107424@mails.zstu.edu.cn (W.Y.)

**Keywords:** vibrationally resolved spectra, nanographenes, DFT calculations

## Abstract

Nanographenes are of increasing importance owing to their potential applications in the photoelectronic field. Meanwhile, recent studies have primarily focused on the pure electronic spectra of nanographenes, which have been found to be inadequate for describing the experimental spectra that contain vibronic progressions. In this study, we focused on the vibronic effect on the electronic transition of a range of chiral nanographenes, especially in the low-energy regions with distinct vibronic progressions, using theoretical calculations. All the calculations were performed at the PBE0-D3(BJ)/def2-TZVP level of theory, adopting both time-dependent and time-independent approaches with Franck–Condon approximation. The resulting calculated curves exhibited good alignment with the experimental data. Notably, for the nanographenes incorporating helicene units, owing to the increasing *π*-extension, the major vibronic modes in the vibrationally resolved spectra differed significantly from those of the primitive helicenes. This investigation suggests that calculations that account for the vibronic effect could have better reproducibility compared with calculations based solely on pure electronic transitions. We anticipate that this study could pave the way for further investigations into optical and chiroptical properties, with a deeper understanding of the vibronic effect, thereby providing theoretical explanations with higher precision on more sophisticated nanographenes.

## 1. Introduction

Nanographenes with atomically precise structures have emerged as pivotal candidates in photoelectronic devices [1,2,3,4], sensing [5,6], as well as single molecular devices [7]. Over the past decades, remarkable progress in synthetic methods has largely facilitated the creation of a rich array of structurally diverse nanographenes that possess unique physical and chemical properties [8,9,10,11,12,13]. Notably, nanographenes with helical [14,15], curved [16,17,18], or twisted configurations are attracting increasing scrutiny [19,20], as the non-planar designs of these molecules not only enhanced the solubility, but also imparted distinctive properties such as chirality [21], dynamic structures [22], or unique packing modes to the molecular skeletons [23]. For example, certain nanographenes reported recently contain one or multiple helicene units [24,25,26,27,28]. these structures, by integrating helicene chirality into the *π*-conjugated backbone, demonstrated pronounced electronic circular dichroism (ECD) and circularly polarized luminescence (CPL) signals, thereby highlighting a notably high CPL brightness (*B*_CPL_) [29,30,31]. Within this research domain, theoretical calculations are gaining increasing importance. Through the utilization of theoretical calculations, a comprehensive exploration of nanographene properties became possible, encompassing the synthetic mechanisms [32], interconversion barriers [33], aromaticity [34], electronic structures [35,36], as well as optical and chiroptical spectra [37,38].

The calculations on electronic spectra, including UV-vis absorption (ABS), emission (EMI), as well as their chiroptical correlatives ECD and CPL, reveal immense significance, as they not only assist in determining the configurations of diastereomers but also provide convincing evidence for assigning absolute configurations between enantiomers [39]. Additionally, they offer detailed insights into molecular excitations, thereby enabling the explanation of the experimental spectra and guiding the design of new nanographene structures with tunable optical properties. Meanwhile, in certain cases of nanographenes or smaller polycyclic arenes, owing to their rigid *π*-skeleton, the electronic spectra were found to be structured, showing several sharp peaks emerging closely. This could be attributed to the vibronic effect on electronic transitions. Taking a polycyclic arene benzo[*ghi*]perylene (**BP**) for example, the experimental ABS spectra showed that the curve contained four peaks at 386, 365, 347, and 330 nm [40]. These peaks could be ascribed to the vibronic progression of transition S_1_. However, according to the theoretical calculation of the pure electronic transition spectrum, only a single band could be simulated, which led to a significant deviation from the experimental result. Larger hexabenzocoronene (**HBC**) also showed similar ABS patterns with structured bands at low-energy regions [41]. Besides, their EMI spectra also exhibited structured curves in a similar manner. Such a phenomenon, caused by the vibronic influence on the electronic excitations, was revealed to be relatively obvious in large *π*-conjugated systems. However, in recent research, most spectral calculations solely focused on electronic excitations, overlooking the vibronic impact on the spectra. In certain cases, substantial disparities between the experimental and calculated spectra could be observed, especially in the low-energy regions, which brought obstacles in analyzing their optical and chiroptical properties.

Recently, a series of investigations exploring the vibrationally resolved spectra have focused on simple *π*-conjugated molecules [42,43,44]. Notably, investigations into several planar compounds have yielded impressive predictions of their optical spectra [40]. Later, in the domain of chiroptical spectra, research on helicenes and their derivatives has also shown strong agreement with both the optical and chiroptical spectra [39,42,43,44]. However, to the best of our knowledge, theoretical calculations regarding the vibrationally resolved optical and chiroptical spectra of chiral nanographenes remain unexplored. In this study, we delve into a series of chiral nanographenes that we have previously synthesized and investigated experimentally (Figure 1) [45]. These compounds showed structured optical and chiroptical electronic spectra, which could serve as valuable objects for studies on vibrationally resolved spectra. The results showed good agreement between the experimental and calculated ABS, EMI, ECD, and CPL spectra after considering the vibronic effect. We anticipate that this study will open up new research avenues in exploring the structure–property relationships of nanographenes with the assistance of theoretical calculations, thereby ultimately guiding the design of innovative nanographene structures with tunable optical and chiroptical properties.

## 2. Results and Discussion

In this study, we have primarily investigated nanographenes **O7H** (the oxidized product of [7] helicene **7H** via an oxidative rearrangement), **O8H** (oxidized from [8] helicene **8H**), **O9H** (oxidized from [9] helicene **9H**), **OO7H** (oxidized from **O7H**), and **OO9H** (oxidized from **O9H**) (Figure 1), which we have synthesized via oxidative cyclo-rearrangement and have studied experimentally previously [45]. From the experimental electronic spectra, these nanographenes show a significant vibronic effect in both ABS and EMI spectra. Besides, the enantiomers of **O8H** and **O9H** also exhibit structured ECD and CPL spectra with vibronic progression. In contrast, our previous theoretical calculations primarily focused on the pure ABS and ECD spectra of these compounds. Thus, a significant discrepancy has been observed between the experimental and calculated spectra, especially in the low-energy region, primarily due to the strong vibronic effect on the electronic transitions [46]. Besides, investigations into the excited states of these molecules have also been conducted. Therefore, we focus on the vibronic impact on the low-energy region of these compounds; we simulate their ABS and EMI spectra for all five compounds and ECD and CPL spectra for the enantiomers of **O8H** and **O9H**, which are involved in the report. In addition, Refs. [5,6,7] helicenes (**5H** to **7H**), as well as **BP**, have also been studied theoretically in this research for comparison.

### 2.1. Pure Electronic Transition of O7H, O8H, and O9H

We initially investigated the phenanthrene-fused helicene derivatives **O7H**, **O8H**, and **O9H**, each containing a [5], [6], or [7] helicene moiety (**5H**, **6H**, and **7H**), respectively. Detailed analysis of the pure electronic ABS spectrum reveals that these nanographenes exhibit the lowest energy transition S_1_ at 2.76, 2.67, and 2.52 eV, respectively (Figure 2). These values closely match the experimental results (Table 1). Besides, this transition also demonstrates a relatively large oscillator strength *f*, ranging from 0.078 to 0.161, consistent with the intense absorption observed in the experimental ABS spectra. Upon the examination of the molecular orbital contribution, we have found that this transition is primarily attributed to the HOMO to LUMO transition with proportions exceeding 95% in all three molecules. The distribution of HOMO and LUMO for **O7H**, **O8H**, and **O9H** are revealed to be similar to **BP** (Figure 3), indicating that their S_1_ transitions are less similar to the primitive helicenes but rather similar to **BP**, where the S_1_ transition is intense with strong vibronic effect. In addition, the second-lowest energy transition S_2_ for these nanographenes is located at 3.05, 2.99, and 2.81 eV, respectively, which is distanced from the first transition. Their oscillator strengths *f* also exhibit extremely low (less than 0.002), indicating that such transitions are almost inhibited. Thus, we suggest that the experimental ABS spectra in the low-energy region (ca. 2.0 to 3.0 eV) are dominantly attributed to the transition S_1_. These electronic transition patterns are much different from their homologs, the primitive helicenes **5H**, **6H**, and **7H** (Appendix A), where the S_1_ transition exhibits low oscillator strength and might be disturbed by the neighboring S_2_ and S_3_ transitions. Obviously, such simple transition patterns of **O7H**, **O8H**, and **O9H** could facilitate the investigation of the vibronic effect on the electronic transitions, and, therefore. we would focus primarily on the transition S_1_ for the ABS spectrum study.

### 2.2. Vibrationally Resolved Electronic Transition of O7H, O8H, and O9H

We first calculated the vibrationally resolved ABS spectra based on a time-dependent (TD) approach for better enhancing the spectral line shape [47]. The calculated ABS spectra using Franck–Condon (FC) approximation and Adiabatic Hessian (AH) model for **O7H**, **O8H**, and **O9H** are presented in Figure 4. To have a better comparison with the experimental spectra, we applied a blueshift to the calculated FC|AH spectra. Specifically, the theoretical spectrum of **O7H** at 0 K exhibits desirable alignment with the experimental curves in the energy range of 2.60 to 3.20 eV, accurately replicating three vibronic peaks at 2.73, 2.90, and 3.08 eV. Taking into account the thermal effect, the spectrum at 300 K shows broadening compared with the curve at 0 K. Besides, the relative height of the peak is also changed. As for larger homologs **O8H** and **O9H**, similar results were observed, with the structured bands reproduced in the calculations at 0 K in low-energy regions. However, for the calculations at 300 K, **O9H** shows severe deviation from the experimental curve.

### 2.3. TI Spectrum and Assignment of the Main Vibronic Bands

To gain a deeper understanding of the vibronic coupling effects and identify the specific vibrational modes responsible for the vibronic progressions, we performed the calculations based on a time-independent (TI) approach, which is grounded in the sum-over-state principle [48,49,50]. The detailed ABS curves and related vibronic transitions are shown in Table 2 and Figure 5. In the TI spectra of **O7H**, the first peak between 2.40 eV and 2.60 eV, primarily originating from the lowest band at 2.46 eV, is associated with the 0-0 transition and combines with a series of S_1_ low-frequency modes: 1 (44 cm^−1^), 3 (88 cm^−1^), 6 (156 cm^−1^), and 22 (443 cm^−1^). These low-frequency modes could be characterized as the movements of CC and CCH out-of-plane bendings (Figure 6a). Notably, mode 3 demonstrates a distinctive motion involving both the shrinking of the **5H** unit and the bending of the **BP** unit. The second peak between 2.60 and 2.80 eV, with noticeably lower intensity, is mainly composed of the fundamentals of S_1_ mode 103 (1502 cm^−1^) in conjunction with the aforementioned low-frequency vibrational modes. These bands typically involve C = C stretching combined with C-H rocking in both the helicene and **BP** units (Figure 6a). Therefore, the interaction between the **BP** and **5H** units likely contributes to the unique vibronic pattern of **O7H**.

The homologs **O8H** and **O9H** also exhibit similar shapes for their ABS curves but with redshifts owing to the extended *π*-conjugation. For instance, **O8H** displayed three calculated vibronic peaks at 2.67, 2.84, and 3.02 eV, which aligned well with the experimental spectrum (at 2.68, 2.85, and 3.01 eV, Figure 4c). Likewise, the calculated TD spectrum of **O9H** has reproduced the vibronic peaks at 2.55 and 2.71 (Figure 4e). In the case of **O8H**, the first peak is predominantly influenced by the 0-0 transition and low-frequency modes, especially modes 1 (34 cm^−1^), 3 (64 cm^−1^), 5 (99 cm^−1^), and 13 (270 cm^−1^), which correspond to the movements of CC and CCH out-of-plane bending (Appendix A). Similarly, the second peak shows a large contribution of frequency mode 108 (1422 cm^−1^), associated with the motion of C = C stretching and C-H rocking (Appendix A). Comparable results were also observed for **O9H**. Notably, mode 3 of **O8H** and mode 3 of **O9H** exhibited a similar vibronic pattern to that of **O7H**.

In the vibrationally resolved EMI spectra, these nanographenes exhibit nearly mirror-image curves compared with the ABS spectra, albeit with slight variations in shape. To illustrate, the calculated TD spectrum of **O7H** at 0 K presents three distinct peaks at 2.64, 2.47, and 2.31 eV, which align almost perfectly with the experimental spectra, where peaks occur at 2.65, 2.49, and 2.33 eV (Figure 4b). However, the spectrum at 300 K shows excessive broadening with a loss of the vibronic information. Similar to the ABS spectra, the first peak with the highest energy is primarily attributed to the 0-0 transition, accompanied by bands related to the S_0_ low-frequency modes 3 (44 cm^−1^), 5 (88 cm^−1^), 6 (156 cm^−1^), and 9 (443 cm^−1^). Notably, these modes are more concentrated on the **BP** framework. The second peak reflects the contribution of vibrational modes 88 (1315 cm^−1^) and 108 (1592 cm^−1^), which could be assigned to C = C stretching and C-H rocking (Figure 6b). In the cases of **O8H** and **O9H**, the calculated EMI curves using the TD approach at 0 K also exhibit sound concordance with the experimental results compared with the curves at 300 K, where the curves at 300 K are found to be excessively broadened. Furthermore, the contributions of vibrational modes, as determined through calculations utilizing the TI approach, have been found to be analogous to those observed in ABS spectra.

### 2.4. Electronic Transition of OO7H and OO9H

In the case of the planar **OO7H**, and its derivative **OO9H** featuring two additional rings that form a **5H** unit, the experimental spectra exhibit more structured shapes compared with previous phenanthrene-fused helicenes, probably owing to their increased structural rigidity. Specifically, for **OO7H**, the experimental spectrum displays three sharp absorption peaks at 3.01, 3.18, and 3.34 eV, while **OO9H** exhibits a similar absorption pattern but with a redshift to 2.75, 2.93, and 3.11 eV. Upon conducting a pure electronic ABS analysis, we found that the lowest transition S_1_ exhibited large oscillator strength (0.1212 for **OO7H** and 0.2254 for **OO9H**) at 3.03 eV for **OO7H** and 2.60 eV for **OO9H**. Besides, transition S_2_, located close to transition S_1_, also exhibits noticeable oscillator strength, but it is much weaker (0.0354 for **OO7H** and 0.0702 for **OO9H**). Concurrently, the transition S_3_ is found to be distanced from both S_1_ and S_2_. Consequently, for the low-energy bands of **OO7H** and **OO9H**, the structured peaks (ranging from 2.80 to 3.20 eV for **OO7H** and from 2.50 to 3.00 eV for **OO9H**) could be mainly contributed by transition S_1_, with a minor contribution from transition S_2_. Thus, we focused on the transition S_1_ for further investigation.

Regarding **OO7H**, the vibrationally resolved calculations on the transition S_1_ show good agreement between the calculated and experimental ABS spectra in the region between 2.70 and 3.30 eV for **OO7H**, reproducing the sharp peaks at 2.84, 3.02, and 3.21 eV in both 0 K and 300 K. Due to its rigid planar molecular structure, the first major peak in the ABS spectrum of **OO7H** is almost entirely dominated by the 0-0 transition. Additionally, there are minor contributions from the S_1_ vibrational modes 12 (312 cm^−1^) and 22 (459 cm^−1^), both of which are in-plane C-C and C-H stretching vibrations (Appendix A). The second peak is contributed by in-plane C = C stretching vibrations, such as vibrational modes 88 (1383 cm^−1^), 90 (1413 cm^−1^), and 112 (1656 cm^−1^) (Appendix A). The EMI spectrum generally mirrors the absorption spectrum but with some differences. The first peak is still mainly contributed by the 0-0 transition, but it is clearly observed that the S_0_ vibrational modes 12 (323 cm^−1^) and 14 (334 cm^−1^), which are also in-plane C-C and C-H stretching vibrations, make significant contributions as well (Appendix A). **OO9H** exhibits characteristics like those of **OO7H**, with the first peaks in both its absorption and emission spectra primarily dominated by the 0-0 transition. However, in the first peaks of the absorption and emission spectra of **OO9H**, the contributing vibrational modes are like the CC and CCH out-of-plane bendings observed in **O7H** (Appendix A). This indicates that the introduction of **5H** affects **OO7H** but does not alter the overall spectral shape and transition properties.

### 2.5. Vibrationally Resolved ECD and CPL of O8H and O9H

Subsequently, we studied the chiroptical properties of the configurationally stable nanographenes **O8H** and **O9H** and calculated their vibrationally resolved ECD and CPL spectra. From the calculated ECD spectra using the TD approach at 0 K, we observed a similar shape of the curves compared with the experimental spectra for both *p*-enantiomers (Figure 7). Notably, in the case of **O8H**, the lowest peak is mainly contributed to 0-0 transition with progressions along S_1_ low-frequency modes 3 (44 cm^−1^), 5 (88 cm^−1^), 6 (156 cm^−1^), and 9 (443 cm^−1^) according to the calculation using the TI approach. This result is similar to the ABS calculations. For the calculated CPL spectra, the results show less accordance with the experimental curves. This might be due to the large noise and broadening in the experimental results.

## 3. Methods

### Computational Details

Density functional theory (DFT) calculations were performed using the Gaussian 09 program [51]. Geometrical optimization calculations were conducted using the PBE0 exchange–correlation functional [52] in combination with the def-TZVP [53] basis set, which employed Grimme’s D3 dispersion approximation [54] (PBE0-D3(BJ)/def-TZVP level) in the gas phase. Harmonic vibration frequency analysis was performed on the optimized geometries to ensure they corresponded to local minima without imaginary frequencies.

Time-dependent density functional theory (TD-DFT) calculations were performed at the PBE0-D3(BJ)/def2-TZVP level in the gas phase. The vibrationally resolved electronic spectra were calculated using the FCclasses 3.0.1 program [55]. Molecular orbitals and hole-electron analyses [56] were generated using the Multiwfn 3.8-dev program [57] and VMD 1.9.3 program [58].

To further investigate the influence of different exchange–correlation functionals, we conducted a benchmark study using the PBE0-D3(BJ), CAM-B3LYP-D3(BJ) [59], M06-2X-D3 [60], and ωB97XD [61] functionals on **BP**; details are provided in the Appendix A.

To directly compare the results with experimental spectra, transition energies and intensities were computed at 0 K and 300 K, respectively, and then convoluted with Gaussian functions with a full width at half-maximum (FWHM) of 0.01 eV. The stick bands of the spectra, depicted as bars in Figure 5 using the TI approach, are expressed as “nx”, where “x” stands for the quanta deposited in the final state (S_1_ for ABS spectra and S_0_ for EMI spectra) of the normal mode *n*.

## 4. Conclusions

In conclusion, we have employed vibrationally resolved spectral analysis to investigate a range of nanographenes that exhibited pronounced vibronic structures in both optical and chiroptical experimental spectroscopy. The calculations were performed using the FC|AH model at the PBE0-D3(BJ)/def2-TZVP level of theory, and both the TD and TI approaches were adopted for the calculations. The calculated results obtained via the TD approach at 0 K exhibited good agreement with the experimental spectra in terms of relative height and width. However, for the calculations at 300 K, owing to the broadening of the spectra, the results were less accurate. Additionally, through the calculations with the TI approach, we discovered that several important vibronic modes contributed largely to the vibrationally resolved spectra of phenanthrene-fused helicenes **O7H**, **O8H**, and **O9H**, which were largely related to the embedded **BP** unit and differed from the primitive helicenes. Unlike the previous computations that solely focused on pure electronic transitions, which led to notable discrepancies, particularly in the low-energy regions of both ABS and ECD spectra, this study underscores the importance and feasibility of examining the vibronic effect in electronic transitions. Consequently, we firmly believe that the introduction of the vibrationally resolved spectral analysis represents a valuable and indispensable tool for future optical and chiroptical studies of nanographenes.

## Figures and Tables

**Figure 1 molecules-29-03999-f001:**
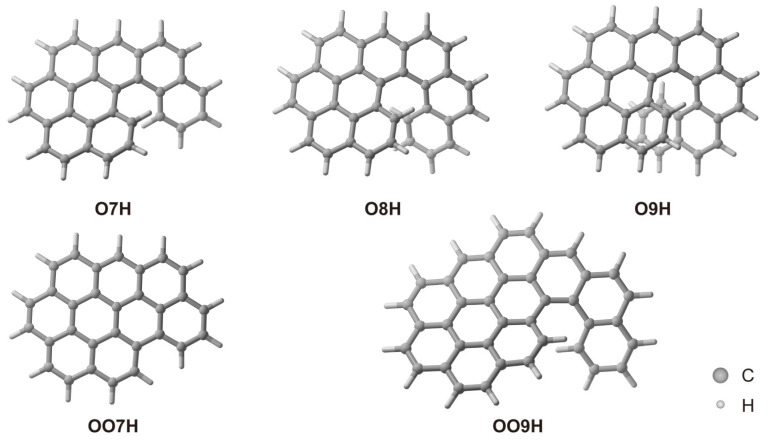
Optimized structure of **O7H**, **O8H**, **O9H**, **OO7H**, and **OO9H**.

**Figure 2 molecules-29-03999-f002:**
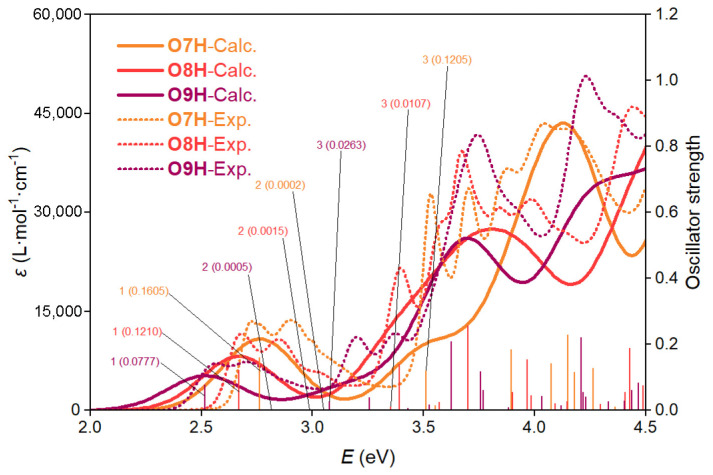
Pure calculated electronic spectra of **O7H**, **O8H**, and **O9H** compared to experimental data from Ref. [45].

**Figure 3 molecules-29-03999-f003:**
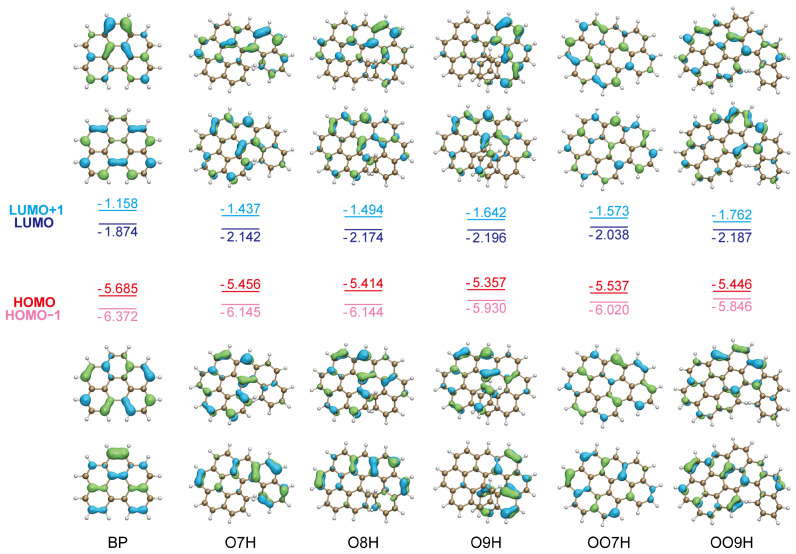
Molecular orbitals and energies (in eV) for **BP**, **O7H**, **O8H**, **O9H**, **OO7H**, and **OO9H**. Calculated using PBE0-D3(BJ)/def2-TZVP level with the isovalue set to 0.05 Å−3.

**Figure 4 molecules-29-03999-f004:**
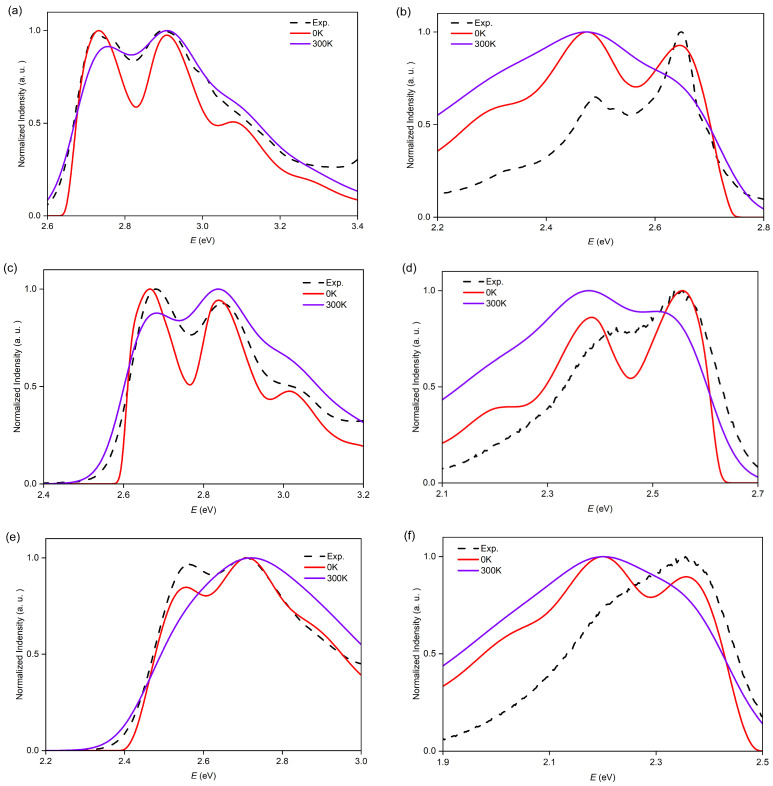
Calculated normalized ABS and EMI spectra of **O7H**, **O8H**, and **O9H** for the transition S1 using the TD approach at 0 K and 300 K, with a comparison to the experimental results from Ref. [45]. (**a**) ABS spectrum of **O7H**, with a blueshift of 0.20 eV; (**b**) EMI spectrum of **O7H**, with a blueshift of 0.27 eV; (**c**) ABS spectrum of **O8H**, with a blueshift of 0.21 eV; (**d**) EMI spectrum of **O8H**, with a blueshift of 0.23 eV; (**e**) ABS spectrum of **O9H**, with a blueshift of 0.23 eV; (**f**) EMI spectrum of **O9H**, with a blueshift of 0.30 eV.

**Figure 5 molecules-29-03999-f005:**
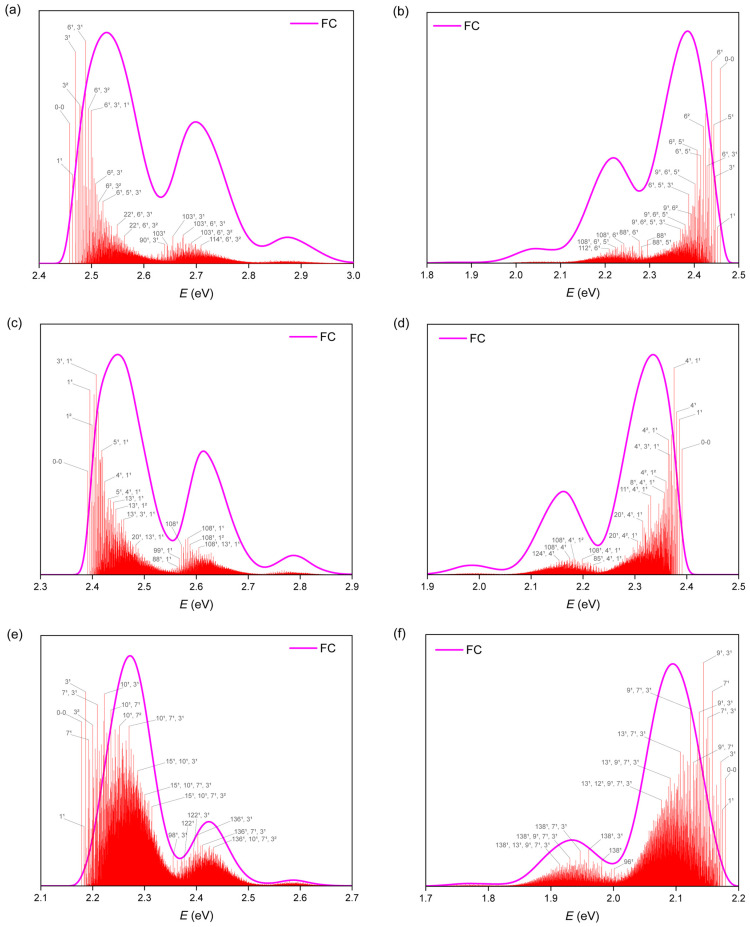
Calculated ABS and EMI spectra of **O7H**, **O8H**, and **O9H** for the transition S_1_ using the TI approach at 0 K, with assignments of the main stick bands of ABS and EMI for **O7H**, **O8H**, and **O9H**. (**a**) ABS spectrum of **O7H**; (**b**) EMI spectrum of **O7H**; (**c**) ABS spectrum of **O8H**; (**d**) EMI spectrum of **O8H**; (**e**) ABS spectrum of **O9H**; (**f**) EMI spectrum of **O9H**.

**Figure 6 molecules-29-03999-f006:**
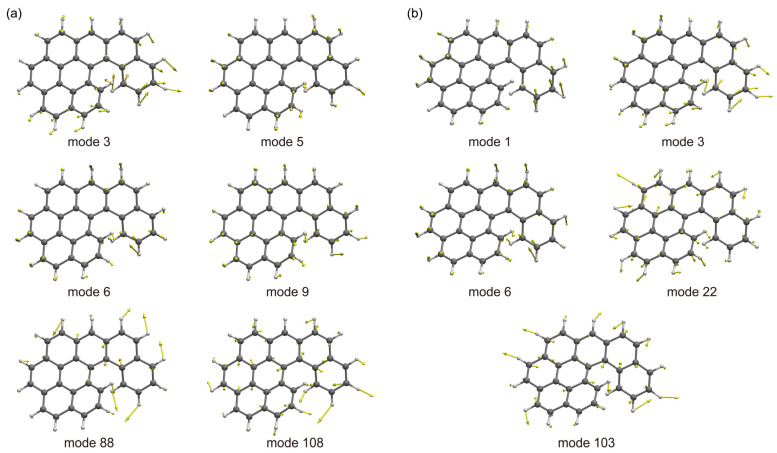
Selected normal modes of vibrations of **O7H**. (**a**) Selected normal modes of vibrations of **O7H** at the S_0_ state; (**b**) selected normal modes of vibrations of **O7H** at the S_1_ state.

**Figure 7 molecules-29-03999-f007:**
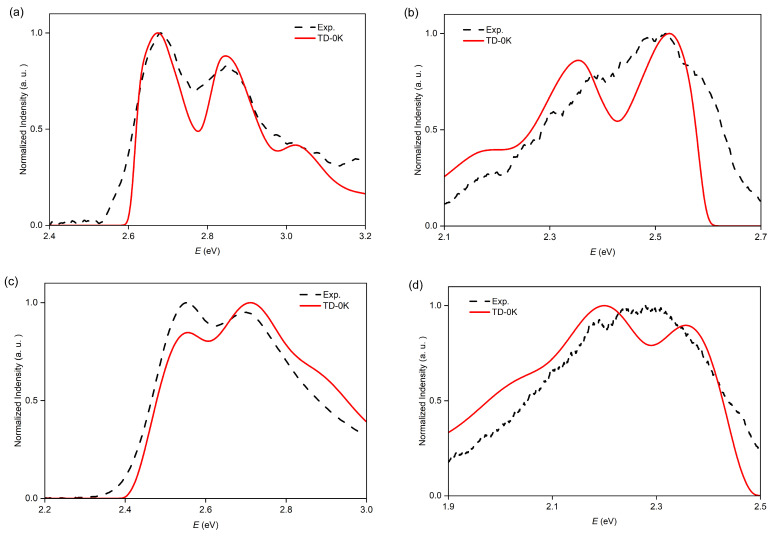
Calculated ECD and CPL spectra of **O8H** and **O9H** for the transition S_1_ using the TD approach at 0 K, with a comparison to the experimental results from Ref. [45]. (**a**) ECD spectrum of **O8H**, with a blueshift of 0.22 eV; (**b**) CPL spectrum of **O8H**, with a blueshift of 0.20 eV; (**c**) ECD spectrum of **O9H**, with a blueshift of 0.23 eV; (**d**) CPL spectrum of **O9H**, with a blueshift of 0.30 eV.

**Table 1 molecules-29-03999-t001:** Absorption and emission information of nanographenes.

	Absorption			Emission			Stokes Shift (nm)
	*E* (eV)	*λ* (nm)	*f*	*E* (eV)	*λ* (nm)	*f*	
**O7H**	2.76	449	0.1605	2.22	558	0.1302	109
**O8H**	2.67	465	0.1210	2.22	559	0.1263	94
**O9H**	2.52	493	0.0777	1.89	657	0.0495	164
**OO7H**	3.03	409	0.1212	2.80	443	0.1883	34
**OO9H**	2.60	477	0.0824	1.93	641	0.0554	164

**Table 2 molecules-29-03999-t002:** Selected vibrational modes relevant to the vibronic structures of **O7H**, **O8H**, and **O9H**.

O7H				O8H				O9H			
S_0_ Mode	ω_0_ (cm^−1^)	S_1_ Mode	ω_1_ (cm^−1^)	S_0_ Mode	ω_0_ (cm^−1^)	S_1_ Mode	ω_1_ (cm^−1^)	S_0_ Mode	ω_0_ (cm^−1^)	S_1_ Mode	ω_1_ (cm^−1^)
3	87	1	44	1	39	1	34	3	56	3	64
5	114	3	88	3	65	3	64	7	112	7	110
6	152	6	156	4	86	5	99	9	168	10	182
9	217	22	443	20	400	13	270	13	244	15	281
88	1315	103	1502	99	1292	108	1422	138	1591	122	1423
108	1592			108	1427					136	1562

## Data Availability

The data are included within the article.

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
