# Peer review of "Theoretical Study on Vibrationally Resolved Electronic Spectra of Chiral Nanographenes"

_molecules, 2024, doi:10.3390/molecules29173999_

Round 1

Reviewer 1 Report

Comments and Suggestions for Authors

In the present work the Authors investigated the vibronic effects on a series of chiral nanographenes through density functional theory-based methods.

In my opinion, the paper shows several shortcomings especially in the discussion of the results. The language is extremely qualitative and at present does not conform to the standards of the Molecules journal. The paper should be revised after the authors have addressed my comments reported here:

1. The abstract needs to be rewritten. It is not clear what the importance of nanographenes is, what are the systems investigated and why they were chosen, what are the challenges to be faced for the study of these systems and the main results. I would also suggest to conclude with a more focused sentence.

2. The introduction is difficult to understand and is a bit confusing.
Please rephrase the sentence at page 2 lines 49-51.

3. The labels used for figure 1 are difficult to understand. I would suggest indicating a numbering for the rings.

4. Please cite some references at page 3 lines 86-87

5. Figure 3 is mentioned before Figure 2

6. It would be better to use the present tense to discuss the results

7. Figure 2. Please indicate the isovalue used to plot the orbitals. 

8. The meaning of this sentence is unclear "For better comparison, the FC|AH spectra have adopted blue-shift with respect to the experimental curves."

9. The labels adopted (BP, 5H, O8H, OO7H and so on..) are very unclear to me. They do not help the reader in understanding the results discussed. 

10. As anticipated in the general comment, the discussion is only qualitative:
"which aligned well with the experimental spectrum."
"which almost aligned with the experimental spectra."
"also exhibited sound concordance with the experimental results compared with the curves at 300 K."

11. The caption of Fig. 6 must be rewritten. I guess that the Authors are referring to "Selected normal modes of vibrations" rather than "Selected frequency modes"

12. The authors should comment on the solvent effects, why did they decide to perform the calculations only in gas phase?

Comments on the Quality of English Language

Moderate editing of English language required

Reviewer 2 Report

Comments and Suggestions for Authors

In this work, the study explores the vibronic effects on the electronic transitions of chiral nanographenes in low-energy regions with DFT calculations. Using the PBE0 functional with D3 dispersion correction in combination with def-TZVP basis, the calculated spectrum aligned well with experimental data. For nanographenes with helicene units, increased π-extension led to significant differences in vibronic modes compared to basic helicenes. The findings could guide further research on optical and chiroptical spectra in more complex nanographenes.

This work can be interesting for the computational chemistry and the industrial community. As such, the proposed article deserves to be published and the Molecules is certainly well targeted. Before the publication, I would like to ask the authors to consider the minor comments below.

1. page 2, Figure 1

Legend for this figure is missing.

2. page 4, Figure 2

Are those HOMO LUMO energies computed at the DFT level? If yes, it should be mentioned in the context that orbital energies computed at the DFT level have errors and the starting point dependence.

3. page 9, Figure 7

In subfigure b and d, compared to experiment results, there are several peaks missed in the calculated spectrum, can the authors provide some discussions on this?

4. Natural transition orbitals should be added in the analysis for excited states, they can be straightforwardly obtained from Gaussian program.

5. In this work, the authors used molecular systems to study the spectrum of nanographenes. Is there any error from not using the periodic boundary condition?

Comments on the Quality of English Language

No major language or grammar problem found.

Round 2

Reviewer 1 Report

Comments and Suggestions for Authors

The revised version of the manuscript can be accepted. The Authors have addressed all the issues raised.